# Preclinical Models of Pancreatic Ductal Adenocarcinoma and Their Utility in Immunotherapy Studies

**DOI:** 10.3390/cancers13030440

**Published:** 2021-01-25

**Authors:** Thao N. D. Pham, Mario A. Shields, Christina Spaulding, Daniel R. Principe, Bo Li, Patrick W. Underwood, Jose G. Trevino, David J. Bentrem, Hidayatullah G. Munshi

**Affiliations:** 1Department of Medicine, Feinberg School of Medicine, Northwestern University, Chicago, IL 60611, USA; mario.shields@northwestern.edu (M.A.S.); christina.spaulding@northwestern.edu (C.S.); 2Jesse Brown VA Medical Center, Chicago, IL 60612, USA; David.Bentrem@nm.org; 3Robert H. Lurie Comprehensive Cancer Center, Chicago, IL 60611, USA; 4Medical Scientist Training Program, University of Illinois, Chicago, IL 60612, USA; principe@illinois.edu; 5Department of Surgery, Feinberg School of Medicine, Northwestern University, Chicago, IL 60611, USA; boli@northwestern.edu; 6Department of Surgery, University of Florida, Gainesville, FL 32611, USA; Patrick.Underwoord@surgery.ufl.edu (P.W.U.); Jose.Trevino@surgery.ufl.edu (J.G.T.)

**Keywords:** immunotherapy, pancreatic cancer, murine models, genetically-engineered mouse models, patient-derived xenografts, organoids, human tumor slice cultures

## Abstract

**Simple Summary:**

Immune checkpoint blockade has provided durable clinical responses in a number of human malignancies, but not in patients with pancreatic cancer. Efforts to understand mechanisms of resistance and increase efficacy of immune checkpoint blockade in pancreatic cancer require the use of appropriate preclinical models in the laboratory. Here, we discuss the benefits, caveats, and potentials for improvement of the most commonly used models, including murine-based and patient-derived models.

**Abstract:**

The advent of immunotherapy has transformed the treatment landscape for several human malignancies. Antibodies against immune checkpoints, such as anti-PD-1/PD-L1 and anti-CTLA-4, demonstrate durable clinical benefits in several cancer types. However, checkpoint blockade has failed to elicit effective anti-tumor responses in pancreatic ductal adenocarcinoma (PDAC), which remains one of the most lethal malignancies with a dismal prognosis. As a result, there are significant efforts to identify novel immune-based combination regimens for PDAC, which are typically first tested in preclinical models. Here, we discuss the utility and limitations of syngeneic and genetically-engineered mouse models that are currently available for testing immunotherapy regimens. We also discuss patient-derived xenograft mouse models, human PDAC organoids, and ex vivo slice cultures of human PDAC tumors that can complement murine models for a more comprehensive approach to predict response and resistance to immunotherapy regimens.

## 1. Introduction

Unlike many other cancers, the incidence of pancreatic ductal adenocarcinoma (PDAC) is increasing [1,2], and the disease is projected to become the second leading cause of cancer-related death by 2030 with a five-year survival rate of ~10% [1]. A number of mutations in key genes such as *KRAS*, *TP53*, *SMAD4*, and *CDKN2A* (p16) have been identified in human PDAC tumors, with activating *KRAS* mutations detected in more than 90% of PDAC tumors [3,4,5]. The discovery of these highly prevalent mutations not only sheds light on our understanding of PDAC progression, but also allows for mechanistic studies investigating novel therapeutic options. Unfortunately, PDAC is significantly more resistant to chemotherapy in comparison to other cancer types, and there are currently limited effective treatments for patients with advanced disease. The combination regimen of 5-fluorouracil, irinotecan and oxaliplatin (FOLFIRINOX) increases median survival from 6.8 months with gemcitabine to 11.1 months [6]. Similarly, the combination of nab-paclitaxel and gemcitabine increases median survival to 8.5 months compared to 6.7 months with gemcitabine [7]. Retrospective studies have shown that targeted therapies based on molecular profiling have the potential to improve survival [8]. Despite these studies, there is still an urgent need to identify and validate new therapies to improve the outcome of PDAC patients.

Among emerging therapeutic targets are immune checkpoints such as PD-1/PD-L1 and CTLA-4. Originally defined as second signals to T cell receptor (TCR) signaling, the engagement and activation of PD-1 and CTLA-4 can effectively suppress T cell proliferation and cytokine production [9]. Early seminal work in the field demonstrating the efficacy of anti-PD-L1/PD-1 [10] and CTLA-4 [11] immune checkpoint inhibitors (ICIs) to enhance anti-tumor immunity culminated in the first FDA approval of anti-CTLA-4 antibody ipilimumab for metastatic melanoma in 2011 and anti-PD-1 antibody pembrolizumab for advanced unresectable melanoma later in 2013. Since then, ICIs have revolutionized the treatment landscape of many solid tumors [12]. However, these therapies have been largely ineffective in most PDAC patients [13,14].

Ample evidence suggests that the lack of ICI efficacy in PDAC is multifactorial but likely caused by a lack of pre-existing, robust antigen-driven T cell immunity [15,16]. While a percentage of patients reportedly have some degree of CD8^+^ T cell infiltration [17,18], the majority of PDAC tumors are classified as immunologically “cold” tumors, characterized by sparse T cell infiltrates (Figure 1) [19,20,21]. This is partly due to the excessive extracellular matrix deposition by cancer-associated fibroblasts (CAFs) [22,23], termed desmoplasia, which creates a physical barrier to infiltrating T cells [24,25]. In addition, PDAC tumors, except for the ~1% that harbor mismatch-repair deficiencies [13,26], have a low mutational burden [3,4,5], associated with lack of cancer-specific epitopes, low immunogenicity, and consequently suboptimal anti-tumor immunity by T cells. It has recently been reported that, even in the presence of neo-epitopes, there is still insignificant T cell response due to scarcity of conventional dendritic cells (cDCs) capable of antigen sampling and migration to tumor-draining lymph nodes to prime T cells [27]. Finally, myeloid cells, particularly tumor-associated macrophages (TAMs, Figure 1), can support tumor angiogenesis and inhibit endogenous T cells via immunosuppressive factors and inhibitory ligands [16,28]. Altogether, the pancreatic tumor immune microenvironment (TIME) present significant challenges to sufficient T cell infiltration and activity, contributing to the clinical absence of ICI efficacy.

Given the clinical success of ICIs in other cancers, there is considerable interest in identifying clinically actionable resistance mechanisms in hopes of overcoming the immunological, biochemical, and physical barriers to T cell infiltration and function in PDAC tumors. However, with the ever-increasing numbers of new immunotherapy agents entering clinical trials, the challenge of how best to evaluate these drugs, either alone or in combination regimens, has become more daunting. Furthermore, establishing and defining the appropriate preclinical models to evaluate certain therapeutic regimens remains a key challenge facing the PDAC treatment landscape [29]. Ideally, the most appropriate preclinical models would not only mirror the biology but also faithfully demonstrate the therapeutic responses seen in patients. Classically, treatment regimens are first tested in murine models of PDAC, which can recapitulate many features of human PDAC tumors. However, these mouse models have several inherent limitations. In this report, we discuss the utility and limitations of multiple PDAC mouse models that are currently available for testing immunotherapy regimens. We also discuss PDAC patient-derived xenografts, PDAC organoids, and ex vivo slice cultures of human PDAC tumors as potential tools that can complement murine models for a more comprehensive guide to predict efficacy as well as resistance to novel immunotherapy regimens.

## 2. Preclinical Immuno-Oncology Models of Pancreatic Cancer

### 2.1. Syngeneic Mouse Models

Syngeneic mouse models of PDAC are developed by introducing immunologically compatible cancerous cells or tissues into mice with an intact immune system, most often by subcutaneous and orthotopic administration. Since subcutaneously implanted PDAC tumors are not in their native organ, namely the pancreas, the biological relevance of subcutaneous models may be limited compared to the orthotopic models. Also, it has been reported that a significant percentage of pancreas-resident macrophages derive from embryonic development and proliferate in situ for expansion [30] and, as such, orthotopically established tumors can display a more fibroinflammatory TIME compared to subcutaneous tumors. However, as subcutaneous models are easier to establish and monitor, they remain a commonly used platform for preclinical drug testing.

#### 2.1.1. Pan02

The Pan02 cell line was derived from a pancreatic tumor developed in a C57BL/6 male mouse following implantation of 3-methylcholanthrene-soaked cotton threads into the pancreas tissue [31]. Pan02 tumors have intrinsically high resistance to a wide range of chemotherapeutic agents and significant metastatic burden [31,32]. Histologically, the Pan02-derived tumors are well-differentiated tumors and do not develop as pronounced a desmoplastic reaction as seen in human PDAC tumors (Figure 2 and [33]). Given the importance of the stroma in tumor immune responses, this may limit the use of allografted Pan02 cells as a model system for immunotherapy, particularly in studies examining crosstalk between stromal and immune cell compartments. In addition, no mutations in *Kras*, *Cdkn2a*, *or Tp53* are detected in Pan02 cells [34]. Lastly, in contrast to human tumors that typically have a low mutational and neoantigen burden [3,4,5], Pan02 tumors demonstrate a very high mutational burden [35], due to its carcinogen-derived tumorigenesis. Recent work by Kinkead et al., identified more than 800 nonsynonymous mutations, with ~20% of these mutations binding MHC I complex with high affinity and eliciting neoantigen-specific T cell responses [35]. This number of neoantigens in Pan02 tumors is noticeably higher than the average number identified in human [36] and spontaneous mouse PDAC tumors [37]. The absence of preexisting T cell immunity despite a high prevalence of naturally occurring neoepitopes in Pan02 tumors is due to several reasons, including inadequate antigen presentation, low frequency of high-quality T cells, and T cells undergoing exhaustion. Indeed, these tumors are often infiltrated with various immunosuppressive cell subsets, including regulatory T cells (Tregs) [38,39,40], myeloid-derived suppressor cells (MDSCs) [40,41] and TAMs [42] (Figure 2), with high expression of inhibitory proteins such as TGFβ [38,43], PD-L1 [40], and CTLA-4 [40]. In contrast, the number of effector T cells is significantly less [40,44], with studies reporting CD8^+^ T cells abundance to be less than 1% [44] (Figure 2). As a result, similar to human tumors, the infiltrating T cells in Pan02 tumors are suppressed, lack effector functions, such as granzyme B secretion [41] and IFN-γ production [40], and often fail to elicit effective anti-tumor responses [39,41,45].

Some studies have reported that subcutaneously implanted Pan02 tumor-bearing mice respond significantly well to anti-PD-1/PD-L1 and anti-CTLA-4 antibody treatment [45,46,47]. Single-agent treatment resulted in marked decrease in immunosuppressive Tregs [47] and increase in the number of effector CD8^+^ T cells with evidence of cytolytic activity (e.g., IFN-γ, granzyme B, and perforin expression) [46]. Their response to PD-L1 therapy can be further enhanced by chemotherapy and radiotherapy [46,48]. Unfortunately, these response rates are significantly higher than the clinical record of ICIs in PDAC patients [13,14], and is possibly related to Pan02 being a carcinogen-induced hypermutated tumors [35]. Exome sequencing of Pan02 tumors identified a number of mutations that were present in a recently reported neoantigen-targeted vaccine capable of inducing antigen-specific T cells within the tumors [35]. Orthotopically established Pan02 tumors are reportedly more resistant to anti-CTLA-4 and PD-1/PD-L1 blockade [49,50] and may better reflect the situation for patients. In these models, anti-CD40 and anti-OX40 were both shown to sensitize tumor-bearing mice to anti-PD-1/PD-L1 ICIs [49,50]. Both CD40 and OX40 are members of the TNF superfamily member. CD40 receptor is expressed on the surface of the antigen-presenting cells (APCs), and its ligation with CD40L ligand modulates their functions and triggers immune responses [51,52]. In the Pan02 mouse model, CD40 agonists have been demonstrated to enhance APC maturation, upregulate Th1 chemokines, leading to increased intratumoral CD8^+^ T cells, and memory T cell expansion [49]. OX40, known as CD134, and its ligand OX40L are also members of the TNF superfamily and expressed on activated T cells [53]. Costimulatory signals from OX40 to conventional T cells support their survival, proliferation, and clonal expansion of effector and memory populations [53]. Interestingly, unlike single-agent treatment with anti-PD-1/PD-L1 or anti-CTLA-4 antibodies, both anti-CD40 and OX40 antibodies as single agents have demonstrated efficacy in Pan02 tumor-bearing mice, in both subcutaneous and orthotopical models [35,49,50]. Together with T-cell–inducing vaccines, OX40 and CD40 synergize with anti-PD-1 antibody to delay tumor progression and improve animal survival [35,54]. The survival benefit was accompanied by significant tumor infiltration by granzyme B-, IFNγ-, and TNFα-secreting effector T cells [35,54].

#### 2.1.2. LSL-Kras^G12D/+^; LSL-Trp53^R172H/+^; Pdx1-Cre (KPC)-Derived Cell Lines

The KPC mouse model (see the Genetically-Engineered Mouse Models (GEMMs) section below) is a well-established genetically-engineered mouse model in which non-immunogenic tumors spontaneously arise [55,56]. Pancreas-specific expression of the mutant Kras and p53 together drive tumorigenesis in immune-competent mice without any exposure to carcinogens [55]. Although the spontaneous KPC tumors recapitulate patient PDAC tumors in terms of tumor development, progression, and histology [55,56], they can take a long time to develop. To circumvent this, different groups have attempted to generate cell lines from these tumors and subsequently implant them back into syngeneic hosts for immunobiology studies. Unlike Pan02 cells, KPC-derived cancer cells express the *Kras* mutation present in the parental tumor. Histologically, allografted tumors developing from implanted KPC cell lines, either subcutaneously or orthotopically, retain a variable degree of desmoplasia (Figure 3). Furthermore, tumors derived from KPC cell lines have a histological appearance and leukocyte complexity similar to the parental tumor arising spontaneously in KPC mice, including a consistently high abundance of immunosuppressive TAMs and a relative lack of T cell infiltration (Figure 3). It has been experimentally demonstrated that the capacity of KPC-derived tumors to exclude T cells is independent of the site of tumor implantation. Tumors implanted subcutaneously, orthotopically, or intrasplenically display heavy infiltration by macrophages and minimal infiltration by effector T cells [17,57]. The capacity to exclude high-quality T cells for immune evasion, often referred to as “immune privilege,” can be a result of number of factors, including insufficient T cell priming and lack of strong immunogenic antigens [58,59]. Indeed, similar to human PDAC tumors, the KPC cell line-derived PDAC tumors demonstrate a low prevalence of missense mutations and immunogenic neoepitopes [37]. Introduction of neoantigens, such as ovalbumin or the more recently described click beetle red [60], can allow intratumoral T cell accumulation and tumor control by T cells [27,37,60], provided there is a sufficient number of functional cDCs [27].

Syngeneic tumors grown from KPC-derived cells are very often used to evaluate ICI efficacy, mostly because they take significantly shorter time to establish while retaining important features of a typical human PDAC tumor as described above. As expected, most KPC-derived tumors have scant intratumoral CD8^+^ T cells, high abundance of TAMs, and respond poorly to single-agent anti-PD-1/PD-L1 and anti-CTLA-4 antibodies [48,61,62]. Recent studies by Li et al., demonstrated that this phenotype is driven strongly by tumor cell-specific expression of lysine demethylase 3A (KDM3A) [63] and CXCL1 (12). Ablation of KDM3A or its downstream signaling targets reprograms the TIME to allow for increased T cell infiltration, reduced myeloid and TAM abundance, and enhanced response to ICIs [63]. Similarly, depletion of CXCL1 promoted PD-1^+^CD8^+^ T cell infiltration and increased responses to immunotherapy regimen in both subcutaneous and orthotopic implanted tumors [17]. This work suggests that perhaps it is the pre-existing intratumoral PD1^+^CD8^+^ T cells, rather than CD8^+^ T cell *per se*, that can predict for sensitivity to ICIs. In fact, previous work in melanoma demonstrated that PD-1 was a marker of tumor-reactive, neoantigen-specific CD8^+^ T cells with potentially high affinity TCRs [64]. Similarly, Kamphorst et al. [65], reported that prior antigen-induced activation or priming of T cells is a pre-requisite for a meaningful response to PD-1 blockade. Significantly, it has been found that orthotopically implanted KPC tumors display increased systemic, tumor-specific T cell exhaustion (with increased PD-1, TIGIT, and LAG-3 protein expression) as they progress over time [60]. It is therefore conceivable that the timing of when the therapy is administered may also be responsible for the varying outcomes seen in the KPC cell line-derived tumors.

KPC-derived tumor model has also been employed to evaluate the effects of immunotherapy-based regimens on tumor cell-extrinsic factors of the TIME. Colony stimulating factor 1 (CSF-1) and its receptor, CSF-1R, regulate various processes in macrophages and other myeloid cells such as migration, proliferation, differentiation, and survival [66]. PDAC patients whose tumors express high levels of CSF-1 are associated with worse prognosis [67,68]. Accordingly, in several preclinical models, inhibition of CSF-1R can lead to preferential depletion of TAMs and reprogram the remaining cells to become more anti-tumorigenic [68,69,70,71]. In mice bearing orthotopic KPC tumors, pharmacologic blockade of CSF-1 reprograms TAMs, reinvigorates CD8^+^ T cells (with increased IFN-γ, granzyme B, Ki67, and CD137 expression), and reduces metastasis [68,71]. The anti-tumorigenic effects of anti-CSF-1R antibody treatment are further potentiated when administered in combination with anti-PD-1 or anti-CTLA-4 antibodies [68,71]. Increased expression of PD-L1 and CTLA-4 on tumor cells, following inhibition of the CSF-1R signaling, may partially account for the observed synergy [68]. Interestingly, it has been reported that anti-CSF-1R therapy had only minimal impact on the intratumoral accumulation of engineered T cells and even interfered with their anti-tumor activity [72]. These findings suggest that endogenous T cells and engineered T cells respond differently to environmental clues. In recent years, targeting TAMs by means of CSF-1/CSF-1R blockade to increase antitumor immunity has generated significant interest [66]. Of note, while a recent phase 2 study of cabiralizumab, an anti-CSF-1R antibody, demonstrated that the agent either alone or in combination with anti-PD-1 antibody is not effective for the treatment of advanced pancreatic cancer (NCT03336216), other anti-CSF-1R antibodies are still being evaluated in earlier phases.

It should be noted that variation of CD8^+^ T cell infiltration levels has been reported for implantable KPC-derived tumors. Li et al. [17], reported high levels of T cells (CD8^+^ and CD3^+^) in a significant proportion (10/24, 41%) of KPC-derived clones. As such, these clones are not ideal to study the immunologically “cold” phenotype typically found in human PDAC. In addition, like most transplantation models, models using the KPC-derived cell lines do not fully mimic the sequential accumulation of mutations seen in human cancers. Furthermore, implantation of cell lines that have undergone extensive passages in vitro can result in unwanted alterations of the TIME that may not be characteristic of human disease [57,73]. As a result, their responses to immunotherapy may vary dramatically from what is seen in human PDAC tumors. For example, while a CD40 agonist-based regimen was found to induce strong T cell-dependent tumor regression in mice bearing subcutaneous KPC tumors [52,74], this treatment was found to be relatively ineffective in PDAC patients, and in those patients with a partial response, the response was through a T cell-independent mechanism of action [75,76].

Together, these findings highlight the potential uses of KPC cell line-derived implantable tumor models in select studies that seek to understand critical aspects by which PDAC tumors regulate T cell exclusion or the capacity of macrophages to alter the TIME. However, efficacy studies of immunotherapy need to be validated in additional models, including GEMMs, and also take into consideration the temporal dynamics of T cell exhaustion during tumor progression.

### 2.2. Genetically-Engineered Mouse Models (GEMMs)

As mentioned earlier, the two most commonly mutated genes in PDAC are the *KRAS* proto-oncogene and the *TP53* tumor suppressor gene [3,4,5]. Alterations in *SMAD4*, *CDKN2A*, and *ARID1A* have also been reported [3,4,5]. Multiple GEMMs of PDAC have been generated to incorporate these alterations specifically in the mouse pancreas, leading to spontaneous tumors that mirror the human disease with high fidelity [77,78]. The most commonly used method to monitor tumor development in these models is ultrasound-guided imaging [56], which can provide a high-resolution view of the developing tumor. It should be noted that most tumors in the KPC GEMM model do not appear apparent by ultrasound imaging until ~12–14 weeks of age.

LSL-Kras^G12D/+^ mice, in which activation of oncogenic Kras allele can be induced, were first described in the early 2000s [79,80]. By crossing Pdx1-Cre mice [81,82] or p48-Cre [83] mice to LSL-Kras^G12D/+^ mice, mutant Kras can be targeted specifically to the pancreas. Bigenic mice are born with normal pancreatic histology and architecture, but develop early pancreatic intraepithelial neoplasia (PanIN) lesions by 8 weeks of age. Few of the KC mice develop PDAC, and the median overall survival of these mice is about 14 months [82]. The LSL-Kras^G12D/+^; Pdx1-Cre (referred to as KC) mice also express the Pdx1-Cre allele in other tissues (such as duodenum, stomach, bile duct) [84]. In contrast, the p48-Cre allele appears to be more specific to the pancreas, though there is some expression in the developing cerebellum and retina [84]. These bigenic mice, especially the KC mice, have been used to study the development of precursor PanIN lesions and strategies to delay tumorigenesis [85,86,87]. The formation of PanIN lesions is accompanied by infiltration of suppressive immune cells, such as MDSCs and macrophages, and less abundantly by cytotoxic CD8^+^ and natural killer T cells [57,78], which mirrors human tumors. It has recently been demonstrated that anti-PD-1 therapy does not provide significant therapeutic benefits in the KC model [88]. One limitation of the KC model is that this model displays incomplete penetrance and long latency, with only a few mice developing metastases [57,82]. To accelerate pancreatic tumor development in the KC mouse model, caerulein, a small oligopeptide that increases secretion of digestive enzymes, has been used [89,90]. Caerulein-induced pancreatic injury leads to increased pancreatic inflammation and acinar-to-ductal metaplasia (ADM), a precursor of PanIN lesions that can progress to PDAC on a Kras mutant background. Additional more aggressive models have been created and combine *Kras* mutations with other genetic alterations, especially in tumor suppressor genes, for instance *Ink4a*/*Arf* and *Smad4*, to accelerate tumor formation and progression [91,92]. Mouse models of PDAC targeting key signaling pathways, such as Hedgehog, Notch, and Wnt, have also been engineered [93]. Studies of T cell responses in these models, however, are scarce.

One of the most commonly studied GEMMs is the LSL-Kras^G12D/+^; LSL-Trp53^R172H/+^; Pdx1-Cre (KPC) model. These mice carry pancreas-specific expression of mutant Kras and p53, which together drive tumorigenesis in immune-competent mice without any exposure to carcinogens [55]. The original KPC model, described by Hingorani et al. [55], was developed on a mixed genetic background (129Sv and C57BL/6). However, most laboratories backcross KPC mice to the C57BL/6J genetic background to achieve an almost pure background without any compromise in tumor penetrance, phenotype, or metastatic propensity. Such a relatively pure background allows the KPC model to be used in adoptive cell transfer studies, such as adoptive transfer of T cells with engineered antigen-specific TCRs [94]. Compared to the KC model, the total leukocyte infiltration is less prominent in the KPC model [57], which is possibly due to the difference in genetics as well as overall survival time between the two models. The KPC model has many features that make it ideal for PDAC biology and therapeutic studies. Histopathological features in KPC mice, such as pronounced desmoplasia (Figure 4), poor vascularity, and high metastatic burden, closely resemble human PDAC tumors [55,56,57]. These tumors are also heavily infiltrated with various immunosuppressive TAMs and much less by effector T cells (Figure 4) [57]. Like human tumors, KPC tumor-bearing mice are mostly refractory to ICIs, including anti-PD-L1/PD-1 and anti-CTLA-4 antibodies [62,95]. As with KPC-derived cell lines, the autochthonous KPC model has a low mutational burden, which accounts for the subsequent scarcity of neoepitopes [17,37] and lack of response to ICIs thereof. It should be noted that the number of neoantigen targets present in the KPC model has been reported to be even lower compared to the human PDAC [35,37].

Notably, the KPC model has provided a basis for several clinical trials. Early studies in the spontaneous KPC model demonstrated that activation of CD40, a receptor broadly expressed by immune cells, can mediate tumor regression in a T cell-independent, macrophage-driven manner [52,75]. This contrasts with findings from implantable KPC models [52,74], and earlier studies in other cancers (lymphoma [96], lung [97]), where T cells were found indispensable for CD40-mediated activity. In the implantable model, treatment with a CD40 agonist was associated with increased infiltration by T cells into the regressing tumors. Depletion of host CD8^+^ and CD4^+^ T cells prior to treatment with the combination of a CD40 agonist and gemcitabine abrogated their efficacy on tumor growth [52], demonstrating a role for T cells in the therapeutic response. In the KPC GEMM model, the combination of a CD40 agonist and gemcitabine induced tumor regression in 30% of tumor-bearing mice [74,75], similar to the objective response rate in patients [75]. Histological analysis of the regressed mouse and human tumors revealed a cellular infiltrate devoid of lymphocytes [75], reiterating CD40-induced anti-tumor activity is independent of T cells. Together, these studies provided the impetus to initiate clinical trials of CD40 agonists in patients with advanced pancreatic cancer [76,98]. In a recent phase 1b study, the combination of a CD40 activating antibody and standard chemotherapy (gemcitabine and nab-paclitaxel) resulted in tumor shrinkage in 20 of 24 patients [98]. Interestingly, immune profiling of patient samples revealed rapid activation of dendritic cells in most patients, suggesting remodeling of the myeloid compartment in response to the treatment regimen [98]. Future studies that evaluate the role of dendritic cells in mediating CD40-induced therapeutic responses will be of particular interest.

The KPC model has also provided insights on mechanisms of TIME-driven immune suppression in PDAC. One identified mechanism involves CXCR4-CXCL12 interaction between CAFs and cancer cells that acts to restrain T cell infiltration [95]. Inhibition of this interaction rapidly induces T cell accumulation and synergizes with anti-PD-1 and anti-PD-L1 antibodies to suppress tumor growth [95]. Based on these findings in the KPC mouse model, a number of synthetic ligands against the CXCR4/CXCL12 axis have been developed, with several reaching clinical trials. Additionally, recent studies have identified TGFβ as a potential immune checkpoint in PDAC [88,99,100]. Principe et al., demonstrated that TGFβ inhibition using the drug galunisertib improved responses to PD-L1 inhibition in KPC mice, particularly in combination with gemcitabine [88,100]. Dual blockade of TGFβ and PD-1 has shown efficacy in early clinical trials, though this has yet to be evaluated in combination with chemotherapy.

While the KPC model has been instrumental in providing rationales for many clinical trials, several combinatorial regimens developed and tested in the KPC mouse model have unfortunately failed to recapitulate preclinical findings in patients. For example, the rationale for combining Hedgehog inhibitors with chemotherapy was initially established in the KPC mouse model, but this combination was unsuccessful in phase 1/2 human trials [101,102]. Similarly, the rationale for targeting the stroma with hyaluronidase was established in the KPC model [103,104]; however, the combination of PEGylated human hyaluronidase and chemotherapy failed to improve survival in a large phase 3 pivotal clinical trial [105]. The exact mechanism(s) underlying this failure may be multifactorial, but some recent studies have highlighted differences in the immune cell composition between KPC and human PDAC tumors, including the relative numbers of TAMs and cytolytic CD8^+^ T cells [17,18,106]. Unlike the myeloid-predominant TIME in KPC mice, T cells (both effector CD8^+^ and regulatory T cells) are found relatively abundant in a portion of human PDAC tumors [17,18,106]. These observations suggest that using the KPC model as the sole model to test preclinical immunotherapy regimens may yield misleading or clinically irrelevant information.

While most PDAC cases occur sporadically, about 4–7% of PDAC patients have a family history of the disease and carry germline mutations in the tumor suppressor genes *BRCA1/2* [107,108]. A recently completed phase 3 clinical trial has found that maintenance therapy with the poly (ADP-ribose) polymerase (PARP) inhibitor olaparib is effective at improving progression-free survival in patients with germline *BRCA*-mutated metastatic PDAC tumors [109]. Two reported mouse models of familial PDAC tumors take advantage of conditional knockout of *Brca2* in the mouse pancreas to recapitulate this subtype. In the model reported by Skoulidis et al. [110], heterozygous loss of *Brca2* is sufficient to promote PDAC driven by Kras^G12D^ (Pdx1-Cre; LSL-KrasG12D; Brca^Tr/Δ11^). In another model, developed by Feldmann and colleagues, biallelic inactivation of *Brca2* promotes PanINs and PDAC, which is further accelerated by *Trp53* mutation (Pdx1-Cre; Brca2^f/f^;LSL-Trp53^R172H^) [111], suggesting a cooperative role of BRCA2 and p53 in pancreatic tumorigenesis. These pancreatic tumors display histological features similar to human PDAC tumors, including abundant desmoplastic stroma, the formation of PanINs preceding tumor development, and metastatic propensities [110,111]. Future studies utilizing mouse models of familial PDAC tumors will be instrumental for evaluating the efficacy of combining PARP inhibitors and immunotherapy regimens.

### 2.3. Patient-Derived Xenografts (PDX) Mouse Models

Like syngeneic mouse models and GEMMs, PDX mouse models have been used with increasing frequency for therapeutic studies in cancer. The PDAC PDX models are developed by surgically implanting freshly resected human PDAC tumor tissue into immuno-deficient mice, typically NOD-SCID gamma (NSG) or NOD/SCIDIL2rgamma^null^ mice [112,113]. The PDAC tumor tissue is implanted subcutaneously or orthotopically in the pancreas, with tumor fragments passaged in subsequent generations [112,113]. The PDAC PDX tumors retain many features of human tumors, including their tumor histology and genetic heterogeneity [112,113]. The PDAC PDX tumors show consistent biological properties and stable phenotypes across multiple passages [112,113]. Clinically, the PDAC PDX models have been used for potential screening platforms for clinical trials, with a good correlation between responses seen in PDX models and in PDAC patients [112,114,115,116].

However, PDX mouse models have some limitations. The establishment of a PDX mouse model from a PDAC patient may take as long as 4–6 months, and the serial passaging of established PDX tumors may take an additional 6–12 months to generate sufficient numbers of tumor-bearing mice for therapeutic studies [112,117]. Since the establishment of PDX tumors requires surgical implantation of fresh tumor tissue, most PDX tumors are developed from patients who have undergone surgical resection of the primary tumor, which is currently only possible in ~15% of PDAC patients [118]. Even then, not all tumors are likely to engraft. It has been found that the more aggressive tumors are more likely to engraft, and the ability to engraft is associated with adverse clinical and pathological features and reduced survival [117]. The genetic heterogeneity may be lost in subsequent passages if it is not represented in the dissected tumor used for the first passage. Also, with the replacement of the human stroma by mouse stroma, there is concern that the PDX tumors may not faithfully represent the fibrotic stroma of the original PDAC tumor [117] (Figure 5). While orthotopic PDAC PDX tumors may better represent the TIME than the subcutaneous tumors, the propagation and expansion of orthotopic tumors for therapeutic studies can be a challenging undertaking.

One of the most significant limitations concerning the use of PDAC PDX tumors for immunotherapeutic studies is that the tumors are often developed and propagated in immunocompromised mice [112,113,117]. PDAC PDX tumors do not retain the donor immune cells, including T cells and macrophages. To facilitate translational immunotherapy studies, there is increasing interest in growing and propagating PDX tumors in mice with some element of the human immune system [119,120,121]. For example, NSG mice have been engrafted with purified human CD34^+^ hematopoietic stem cells obtained from bone marrow, umbilical blood, or fetal liver [119,120,121]. These humanized PDX models have been successfully used to examine the response to ICIs for melanoma, hepatocellular carcinoma, and lung cancer [122,123]. However, to our knowledge, there are currently no published reports on the use of humanized PDX models to test immunotherapy regimens in PDAC. Finally, it is noteworthy that since the immune system of the humanized PDX PDAC mouse model is not derived from the same patient, responses seen in the model may fail to predict clinical responses expected or observed in the donor patient.

### 2.4. Human PDAC Organzoid Cultures

To address some of the limitations of PDX models, there is increasing interest in employing organoid cultures for therapeutic studies [124,125]. PDAC organoids can be established from surgical resection specimens and fine needle biopsy specimens at high rates. This enables establishing PDAC organoids from both resected tumor specimens and metastatic disease [126,127]. Studies have shown that PDAC organoids can be established with a success rate of 65–75% [126,128,129]. Importantly, PDAC organoids maintain tumor heterogeneity even after numerous passages and retain the mutational complexity seen in human PDAC tumors [126,128,129]. These organoids recapitulate many features of human disease when orthotopically transplanted into mice, such as the extensive stromal reaction seen in human PDAC tumors [126,128,129].

Even though the establishment and maintenance of organoids is costly and is labor- and time-intensive, many investigators have successfully established biobanks to conduct basic and translational research. Using a library of 66 human PDAC organoids, the Tuveson group has established chemotherapy-sensitivity profiles for each organoid in a clinically meaningful timeline [128]. Importantly, these chemotherapy-sensitivity profiles are reproducible in patients [128]. Similarly, the Clevers group screened 76 therapeutic agents against a panel of 30 PDAC organoids and identified unique drug-sensitivity profiles for individual organoid lines [130]. For instance, loss of methylthioadenosine phosphorylase (MTAP) results in sensitivity to inhibitors targeting protein arginine methyltransferase 5 (PRMT5) [130]. Finally, organoids have also been used to identify and validate mechanisms of therapy resistance. For example, activation of ERBB2 and ERBB3 following treatment with MEK and AKT inhibitors confers resistance to the treatment [131]. As a result, the combination of pan-ERBB inhibitors and MEK inhibitors demonstrates higher anti-tumor activity than either agent against PDAC organoids in vitro as well as in orthotopically implanted organoids [131].

However, the PDAC organoids grown in culture lack a TIME (including fibroblasts and immune cells), limiting their use for immuno-oncology studies. As a result, there are efforts to generate co-culture organoids with other cell types. Pancreatic tumors are associated with pronounced stroma containing CAFs and pancreatic stellate cells [22,23]. The Tuveson group has established co-culture systems to incorporate stellate cells and fibroblasts and, in the process, identified distinct CAF populations that are induced by different ligands [132,133]. These findings provide direct evidence for CAF heterogeneity in PDAC tumor biology with implications for disease etiology and therapeutic development. Other groups have also successfully incorporated T cells and fibroblasts to generate a triple co-culture model of PDAC organoids [129,134].

The utility of organoids in immuno-oncology studies has recently been tested. Using alternative culture conditions from that established by the Tuveson and Clevers groups, the Kuo group was able to propagate organoids with immune cells and fibroblasts for several weeks [129]. Their air-liquid interface (ALI) cultures display T cell clonal diversity that mirror the T cell diversity seen in the patient’s peripheral blood [129]. The new, optimized protocol allows for successful establishment of ALI cultures in 11/17 (65%) PDAC tumors [129]. Significantly, they have employed these cultures to test ICIs in several different tumor types, including melanoma, renal cell cancer, and non-small cell lung cancer [129]. While their recent report did not include PDAC organoids, the ALI organoid cultures have the potential for preclinical evaluation of ICIs for PDAC. It should be noted that while the organoid cultures are useful for in vitro testing of anti-tumor responses, they cannot be used to test the safety of immunotherapy drugs.

### 2.5. Ex Vivo Slice Cultures of Human PDAC Tumors

In addition to organoid cultures, ex vivo tissue slices of human PDAC tumors have also been utilized to complement mouse models in therapeutic studies. The tumor slice cultures, with sections of 250–400 μM in thickness precisely cut with a vibrating microtome, maintain tumor histology, morphology, proliferation, and viability for up to 7 days [18,135,136]. The integrity of the PDAC TIME is maintained, with stromal myofibroblasts surviving slice cultures [18,135,136]. Similarly, immune cells, such as CD8^+^ T lymphocytes and CD68^+^ macrophages are preserved during culture (Figure 6), with most immunological proteins staying stable and detectable during the culture process [18,135,136]. Finally, slice cultures appear to stay metabolically active during the culture time [136]. As the closest surrogate to the parent carcinoma to date, this culture system holds great potential as a drug sensitivity testing system for the personalized treatment of PDAC.

In addition to several reports on chemotherapy and targeted therapy [135,137], the PDAC slice cultures have lately been used to evaluate and visualize immune cell response to ICIs. Treatment of slice cultures with the CXCR4 inhibitor AMD3100 and anti-PD-1 antibody enhanced the effector function of CD8^+^ T cells [18]. The authors also performed live microscopy to demonstrate that CXCR4 inhibition allowed T cells to migrate through the stromal microenvironment and reach tumor cells [18]. These findings successfully recapitulated previous study by Feig C et al. [95], in which KPC GEMM responded to AMD3100 with increased intratumoral infiltration by T cells, highlighting the utility of this ex vivo platform to reproduce meaningful data in a significantly shorter time. By allowing incorporation of the patients’ immune cells isolated from their peripheral blood [18,135], slice culture model has the potential to examine effects of immune-based regimens on immune cell infiltrates, other than CD8^+^ T cells.

It is important to note some of the limitations of PDAC slice cultures for immuno-oncology studies. Like organoids, slice cultures cannot be used to evaluate therapeutics safety. In addition, with the current culture conditions, the PDAC slices remain viable for approximately one week [135,136]. Since some of the immune responses may take longer than a week to manifest, the current slice culture conditions do not allow for long-term treatment and thus may limit the utility of PDAC slice cultures in immuno-oncology studies. Since there is significant variability in term of immune cells distribution within human PDAC tumors [138], response to treatment may vary significantly depending on the sections used for slice cultures. Also, as mentioned earlier, there is immune cell heterogeneity in the PDAC tumors between patients [17,18,106], which may require PDAC tumors from a sufficient numbers of patients to demonstrate response to therapy. However, a positive demonstration of response in PDAC slice cultures from multiple patients could provide a strong rationale to advance a particular regimen to additional validating in vivo studies. A more significant limitation is that while it is possible to use peripheral blood to isolate and incorporate T cells in ex vivo cultures [18,135], often the immune composition and response in PDAC slice cultures are limited to the tumor-infiltrating immune cells that are already present in the tumors [18,135]. It has been shown that a good proportion of human PDAC tumors do not have sufficient numbers of infiltrating CD8^+^ T cells [19,20,21], and the slice cultures do not easily allow for evaluation of primed immune cells entering the tumor from the circulation as a result of treatment with the different therapies.

## 3. Conclusions

Given the increasing interest in developing and evaluating new immunotherapy regimens for PDAC, it is important to be aware of the limitations of different preclinical models (Table 1). Although cell line-based models have many advantages, these models often do not faithfully replicate the stromal microenvironment or the immune cell composition seen in human tumors. While GEMMs demonstrate low mutational burden and histopathological features similar to human tumors, the differences in their immune cell composition compared to human tumors represent a significant limitation. In contrast, the PDX models account for the tumor heterogeneity seen in human PDAC tumors and allow for testing of immunotherapy regimens in the humanized mouse. However, the reconstituted immune system is often not derived from the patient from whom the PDAC PDX tumor was initially established. Thus, advances in the humanized mouse model system may increase the utility of PDX models for testing of immunotherapy regimens, not only for PDAC but also for other cancer types. Similarly, the ability to incorporate a more complete repertoire of immune and stromal cells in organoids may also increase their utility in evaluating immunotherapy regimens. Finally, modifications in slice culture protocols that allow for extended in vitro maintenance and incorporation of primed immune cells will significantly improve the model. Overall, efforts to combine multiple models for preclinical testing and validating ICIs have the potential to identify immunotherapy regimens that are more likely to translate into successful clinical trials for patients with PDAC.

## Figures and Tables

**Figure 1 cancers-13-00440-f001:**
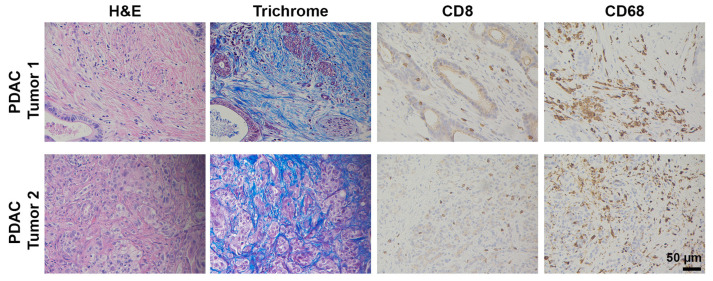
Human PDAC tumors display pronounced desmoplasia macrophages, but lack CD8^+^ T cells. Human PDAC tumors were H&E or trichrome stained or stained by immunohistochemistry for CD8^+^ T cells (Abcam, #4055) and the macrophage marker CD68 (Cell Signaling, #76437). Magnification, 20×.

**Figure 2 cancers-13-00440-f002:**
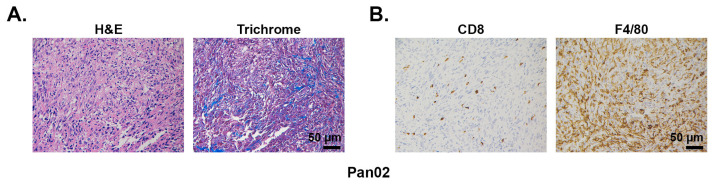
Desmoplasia, CD8^+^ T cells, and macrophage infiltration in mouse Pan02 tumors. Pan02 tumors developing subcutaneously in C57BL/6J mice were H&E and trichrome stained (**A**) or stained by immunohistochemistry for CD8^+^ T cells and the macrophage marker F4/80 (**B**). Antibodies used: CD8^+^ (Cell Signaling, #98941) and F4/80 (Cell Signaling, #70076). Magnification, 20×.

**Figure 3 cancers-13-00440-f003:**
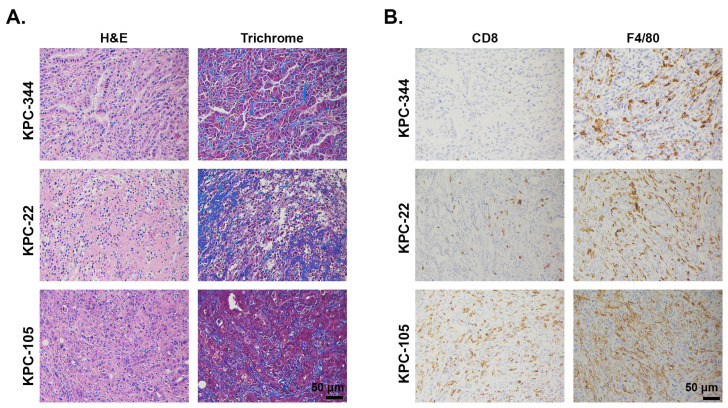
Desmoplasia, CD8^+^ T cells, and macrophage infiltration in mouse implantable KPC tumors. KPC cell lines (KPC-344, KPC-22, KPC-105) established from KPC spontaneous mouse model were injected subcutaneously in C57BL/6J mice. The tumors were H&E or trichrome stained (**A**) or stained by immunohistochemistry for CD8^+^ T cells and the macrophage marker F4/80 (**B**). Antibodies used: CD8 (Cell Signaling, #98941) and F4/80 (Cell Signaling, #70076). Magnification, 20×.

**Figure 4 cancers-13-00440-f004:**
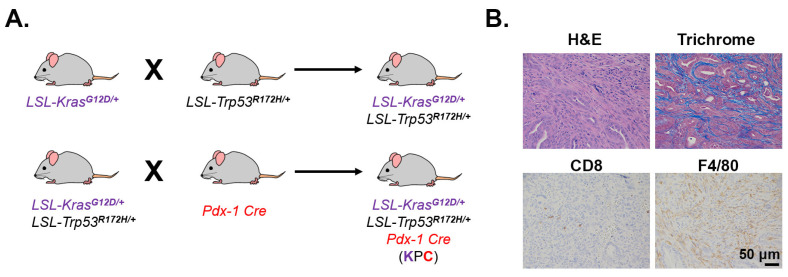
Desmoplasia, CD8^+^ T cells, and macrophage infiltration in GEMM KPC tumors. (**A**). Breeding scheme to generate the genetically engineered KPC mouse model. LSL, LoxP-Stop-LoxP; Pdx-1, pancreatic and duodenal homeobox-1. (**B**). A PDAC tumor in a 6-month old KPC mouse was H&E or trichrome stained (10×) or stained by immunohistochemistry for CD8^+^ T cells and the macrophage marker F4/80 (20×). Antibodies used: CD8 (Cell Signaling, #98941) and F4/80 (Cell Signaling, #70076).

**Figure 5 cancers-13-00440-f005:**
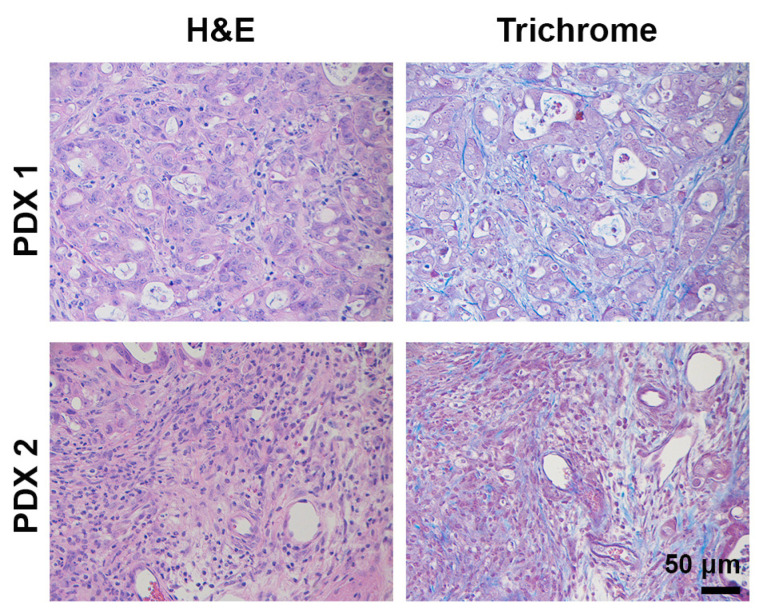
Desmoplasia and immune infiltration in PDAC PDX tumors. Tumors from chemo-naïve PDAC patients were established in immunodeficient mice, and passage #2 tumors were H&E and trichrome stained. Magnification, 20×.

**Figure 6 cancers-13-00440-f006:**
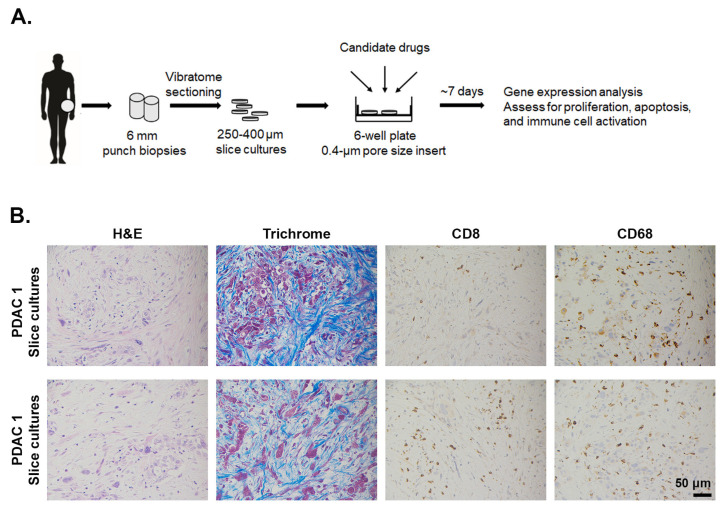
Desmoplasia, CD8^+^ T cells, and macrophage infiltration in ex vivo PDAC slice culture. (**A**). The procedure to generate and maintain tumor slice culture in vivo. 6 mm punch biopsies collected freshly from patients are sectioned into slices that are 250–400 μm thick. The slices are then placed on membrane inserts in 6-well plates to create an air–liquid interface. During the course of 5–7 days, tumor slices are subject to different drug treatments. (**B**). After 5 days in culture, the tumor slices were H&E or trichrome stained or stained by immunohistochemistry for CD8^+^ T cells (Abcam, #4055) and the macrophage marker CD68 (Cell Signaling, #76437). Magnification 20×.

**Table 1 cancers-13-00440-t001:** Advantages and drawbacks of different preclinical models of PDAC for immunobiology studies.

Model	Principle	Benefits	Caveats
Syngeneic Pan02 cell line	Established mouse tumor cell lines transplanted (subcutaneous or orthotopic) into immune-competent C57BL/6J mice	Presence of an intact immune system Tumor microenvironment is strongly immunosuppressive and scarce in effector CD8+ T-cells Tumors are intrinsically resistant to a wide range of chemotherapeutics Subcutaneously implanted tumors can be easily monitored and measured	Limited desmoplastic reaction High mutational load since derived from a carcinogen-induced tumor Lack Kras mutation Higher response to anti-PD-1 and anti-CTLA-4 antibodies than what is typically seen in human tumors
Syngeneic KPC-derived cell lines	Established mouse tumor cell lines transplanted (subcutaneous or orthotopic) into immune-competent C57BL/6J mice	Presence of an intact immune system Tumor microenvironment is strongly immunosuppressive Tumors display desmoplastic reaction Tumors demonstrate low incidence of missense mutations and lack of neoepitopes, similar to human tumors Subcutaneously implanted tumors can be easily monitored and measured	Tumors exhibit variable abundance of intratumoral CD8+ T-cells Tumors do not fully mimic the sequential accumulation of mutations seen in human cancers Tumors derived from late passages have unpredictable and unexpected alterations in the microenvironment The tumor microenvironment of subcutaneous tumor may not be reflective that of orthotopic tumors
Genetically-engineered KPC mouse model	Concurrent activation of Kras^G12D^ and loss of p53 in the pancreas to allow for spontaneous tumor development	Presence of an intact immune system Tumor microenvironment is strongly immunosuppressive Histopathological features closely resemble human tumors: cellular morphology, poor vascularity, pronounced desmoplasia, and metastatic spread Tumors demonstrate low incidence of missense mutations and lack of neoepitopes	Tumors exhibit variable abundance of intratumoral CD8+ T-cells Tumors can have higher abundance of myeloid cells than what is seen in human tumors Low throughput and high investment
Patient-Derived Xenograft (PDX) Mouse Models	Human PDAC tumors implanted in immune-deficient host	Early passaged PDX tumors preserve the tumor histology and the genetic heterogeneity of host patients PDX tumors show consistent biological properties and stable phenotypes across multiple passages PDX tumors show therapeutic responses similar to what seen in patients	Tumor-bearing mice lack an intact immune system Surgical implantation is required with low engraftment rates PDX tumors do not faithfully represent the stroma of the original PDAC tumor Heterogeneity may be lost in subsequent serial passages Propagation and expansion of orthotopic tumors can be challenging
Organoid cultures	Tumor cells isolated from human tumors allowed to grow into 3D structures in specialized growth media	PDAC organoids maintain tumor heterogeneity and mutational complexity after serial passages PDAC organoids retain the extensive stromal reaction when orthotopically implanted in mice PDAC organoids can be from both resected tumor specimens and metastatic disease with high engraftment rates PDAC organoids show chemotherapeutic responses similar to what seen in patients	Establishment and maintenance of organoids is labor- and time-intensive PDAC organoids grown in culture often lack a tumor microenvironment, including fibroblasts and immune cells
Ex vivo slice cultures	250 µm to 400 µm thick sections from human PDAC tumors maintained and cultured in vitro	Slice cultures maintain tumor morphology, proliferation, and viability ex vivo The original integrity of the PDAC tumor microenvironment is maintained, with stromal cells surviving culture conditions Slice cultures retain immune infiltrates from the host patients	Current slice culture conditions do not allow for long-term treatment (only up to 1 week) Significant heterogeneity in the distribution of immune cells are seen with slices dissected from different regions Current slice cultures do not allow for evaluation of primed immune cells entering the tumor from the circulation

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
