# Peer review of "Preclinical Models of Pancreatic Ductal Adenocarcinoma and Their Utility in Immunotherapy Studies"

_cancers, 2021, doi:10.3390/cancers13030440_

Round 1

Reviewer 1 Report

Thao N.D. Pham et al in this review systemically described the pre-clinical Immuno-Oncology Models of pancreatic ductal adenocarcinoma (PDAC). The authors described Immuno-Oncology Models of pancreatic cancer models including Syngeneic Mouse Models, Genetically-engineered mouse models (GEMMs), Patient-derived Xenografts (PDX) Mouse Models, Human PDAC organoid cultures, Ex vivo slice cultures of human PDAC tumors and compared their clinical features with the human PDAC and emphasized their utility and limitations for testing immunotherapy regimens. The review is well organized and written. The authors may speculate the future best models for the PDAC.

Author Response

We thank the reviewer for the suggestion. As presented in the manuscript, the currently available preclinical models have their own benefits and drawbacks. We therefore have again emphasized in the conclusion that a combination of models is necessary to study the biology and therapeutic response of immunotherapy in PDAC. Please see lines 574-577.

Reviewer 2 Report

Authors have described the benefits, caveats, and potentials for improvement of the most commonly used models, including mouse and humans. The review describes the recent strategies and therapeutic insights in pancreatic cancer. Following minor points can be addressed before publication.

Comments: KPC mice should be given in full name in page 5, where its initially described (though details have been given later).

The current therapeutic approaches in PDAC and the failures should be presented to give a glimpse of the deadly disease.

Author Response

Comments:

  1. KPC mice should be given in full name in page 5, where its initially described (though details have been given later).

Response: We have now included the full name for the KPC abbreviation in page 5. Please see line 181.

  1. The current therapeutic approaches in PDAC and the failures should be presented to give a glimpse of the deadly disease.

Response: We thank the reviewer for the suggestion. We have now included a discussion under the Introduction about the limitations of the current approved therapeutic approaches for PDAC. The new information is highlighted in lines 48-55.

Reviewer 3 Report

In this review the authors discuss the benefits and caveats for the commonly used models of pancreatic cancer. The information in this review will be useful for researchers interested in identifying an appropriate model to study immunotherapies in PDAC.

However, it would benefit if the authors reorganize the review.

1) in the introduction the authors should mention as to what would be an ideal model for studying ICIs in PDAC.

2) The authors start by discussing the syngenic Pan02 and KPC cell line based mouse models and then move to GEMMs and while discussing the GEMMS they again discuss the KPC model which is very confusing. As a suggestion they can reorganize it in such a way that they explain the GEMMs first and then deal with the Syngenic cell line based models.

3) importantly, the authors should conclude each section with a short summary of the pro and cons of that model in the context of studying immunotherapies. This could improve the continuity across the sections.

Author Response

  1. In the introduction the authors should mention as to what would be an ideal model for studying ICIs in PDAC.

Response: We have now included our assessment as to what is an ideal model for studying ICIs in PDAC. As presented in the manuscript, many currently available preclinical models have both advantages and disadvantages. We speculate that an ideal model would be one that recapitulates responses seen in patients, not only to ICIs alone but also to chemotherapeutics and targeted therapies. Please see lines 96-98.

  1. The authors start by discussing the syngenic Pan02 and KPC cell line based mouse models and then move to GEMMs and while discussing the GEMMS they again discuss the KPC model which is very confusing. As a suggestion they can reorganize it in such a way that they explain the GEMMs first and then deal with the Syngenic cell line based models.

Response: We thank the reviewer for the suggestion. The current structural organization of the manuscript is an attempt to describe these preclinical models according to their complexity in terms of establishing and maintaining. For that reason, syngeneic models were placed before GEMMs. To improve the flow, we have now included a brief description of the KPC GEMM model under the syngeneic section. Please see highlighted changes in lines 182-189.

  1. Importantly, the authors should conclude each section with a short summary of the pro and cons of that model in the context of studying immunotherapies. This could improve the continuity across the sections.

Response: We have systematically summarized the pros and cons of the preclinical models in Table 1. We think that by placing next to each other the pros and cons of all the models included in the work will help with the continuity and overall flow of the manuscript. The table is on page 16-17.